# VITA-VLA: Efficiently Teaching Vision-Language Models to Act via Action Expert Distillation

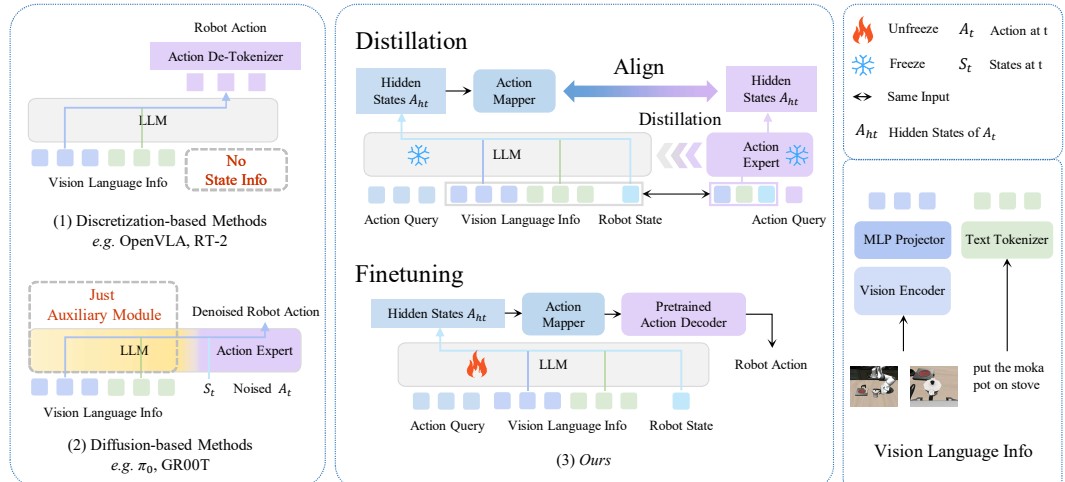

Figure 1: **Overview of mainstream VLA architectures.** (1) Discretization-based methods map vision and language features into action tokens via LLM, but ignore robot state—an essential signal of physical dynamics—making action prediction less effective. (2) Diffusion-based approaches extract vision and language features with a VLM but pass them to a separate action expert for denoising, reducing the VLM to a large feature extractor and limiting its overall capability in action modeling. (3) Our model distills knowledge from a small action model while largely preserving the VLM structure. By integrating robot state through a lightweight encoder and introducing an action token to fuse vision, language, and state, it enables the VLM to actively participate in action modeling rather than only serving as a feature extractor, thereby better leveraging its modeling capabilities.

## Abstract

Vision-Language Action (VLA) models significantly advance robotic manipulation by leveraging the strong perception capabilities of pretrained vision-language models (VLMs). By integrating action modules into these pretrained models, VLA methods exhibit improved generalization and robustness. However, training them end-to-end is costly, as modeling action distributions typically requires massive datasets and heavy computation. In this work, we propose a simple yet effective distillation-based framework that equips VLMs with action-execution capability by transferring knowledge from pretrained small action models. Our architecture retains the original VLM structure, adding only an action token and a state encoder to incorporate physical inputs, as illustrated in Figure 1. To distill action knowledge, we adopt a two-stage training strategy. First, we perform lightweight alignment by mapping VLM hidden states into the action space of the small action model, enabling effective reuse of its pretrained action decoder and avoiding expensive end-to-end pretraining. This also facilitates better transfer of action modeling capabilities to the VLM. Second, we selectively fine-tune the language model, state encoder, and action modules, enabling the system to integrate multimodal inputs with precise action generation. Specifically, the action token provides the VLM with a direct handle for predicting future actions, while the state encoder allows the model to incorporate robot dynamics not captured by

vision alone (see Figure 2). This design yields substantial efficiency gains over training large VLA models from scratch. Compared with previous state-of-the-art methods, our method achieves 97.3% average success rate on LIBERO (11.8% improvement), 93.5% on LIBERO-LONG (24.5% improvement), 92.5% first task success rate on CALVIN ABC-D (4.1% improvement). In real-world experiments across five manipulation tasks, our method consistently outperforms the teacher model Seer, achieving 82.0% average success rate (17% improvement). These results demonstrate that action distillation effectively enables VLMs to generate precise, executable actions while substantially reducing training costs.

# 1 INTRODUCTION

Traditional small action models typically have limited parameters, are trained in fixed environments, and excel at executing simple predefined tasks (Wu et al., 2023; Black et al., 2023; Tian et al., 2024). However, these models often struggle to generalize to dynamic or perturbed environments, which restricts their effectiveness in real-world scenarios. In contrast, VLMs demonstrate remarkable visual comprehension and instruction-following capabilities, exhibiting strong generalization performance across a wide range of tasks. This motivates growing interest in combining VLMs with action models, leading to the development of various VLA models, such as the RT series (Brohan et al., 2022; Zitkovich et al., 2023), OpenVLA (Kim et al., 2024), GR00T (Bjorck et al., 2025), $\pi_0$ (Black et al., 2024), Octo (Team et al., 2024), 3D-VLA (Zhen et al., 2024), and others (Cui et al., 2025; Zhao et al., 2025; Qu et al., 2025). These models can be broadly categorized into two main approaches, as illustrated in Figure 1. The first category, exemplified by OpenVLA and the RT series, adopts a discretization-based approach. These models transform continuous actions into discrete tokens by partitioning the action space into fixed intervals. Visual and language features are then mapped to these action tokens via a Large Language Model (LLM), which are subsequently detokenized into executable actions. The second category, represented by GR00T and $\pi_0$, follows a diffusion-based design. Vision-language features are first extracted by a pretrained VLM and then injected into the action expert through attention mechanisms. The action expert then iteratively refines the noised action representations conditioned on the current state, noised actions, and vision-language features, to generate the final executable actions. While both categories have demonstrated promising performance, they exhibit notable limitations. Discretization-based approaches often omit robot state information—a crucial signal for modeling physical dynamics—which can hinder the accuracy of action prediction. Meanwhile, diffusion-based methods typically leverage the VLM solely as a feature extractor, reducing it to a static encoder and underutilizing its potential for end-to-end action modeling. Furthermore, despite being trained with extensive computational resources and large-scale, high-quality data, they still lag behind smaller, task-specific models on embodied benchmarks like CALVIN (Mees et al., 2022) and LIBERO (Liu et al., 2023a).

To address these limitations, we propose a new VLA architecture and a two-stage training framework that equips pretrained VLMs with action-generation capability via knowledge distillation from small action models(Figure 1 (3)). This approach removes the need for expensive end-to-end pretraining on large embodied datasets and achieves strong performance across simulation benchmarks and real-world experiment. Our method is built upon the VITA-1.5 architecture, which is based on the well-known and widely-used LLaVA architecture (Liu et al., 2023b). We extend this backbone with two components—a state encoder and an action token—so that it can fuse visual, language, and state inputs to generate executable robot actions, as illustrated in Figure 2. To efficiently equip the VLM with action capabilities, we introduce a simple yet effective two-stage training framework centered on knowledge distillation, as illustrated in Figure 3. In the first stage, we perform lightweight alignment by projecting the VLM's hidden features into the action space of a pretrained small action model. This alignment process explicitly distills the action-generation policy from the small action model into the VLM. By doing so, we can reuse the action decoder of the small action model, avoiding the need for costly end-to-end pretraining from scratch. In the second stage, we fine-tune specific components of the system, including the language model, state encoder, and action modules. This selective fine-tuning helps to better integrate multimodal signals, leading to accurate action predictions. This two-stage approach not only enhances the VLM's ability to model complex robotic behaviors but also reduces training cost, while retaining strong generalization.

We validate the effectiveness of our approach on both simulation benchmarks and real-world robotic tasks. On the LIBERO benchmark, our two-stage training strategy achieves a **97.3%** average suc-

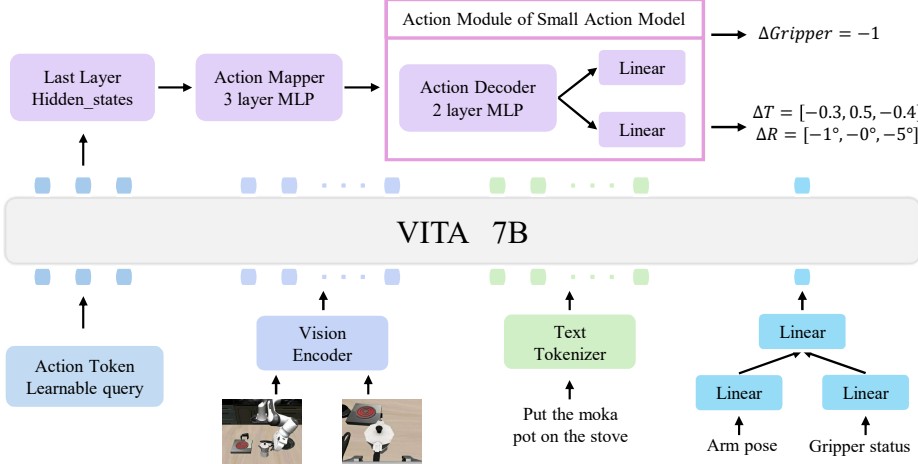

Figure 2: **Model Architecture.** Our model is build upon VITA1.5-7B (Fu et al., 2025), taking images, instructions, action tokens, and state information as inputs to generate executable actions. The visual and textual information is input into the VLM. The action token acts as a learnable query, while the robot state is encoded into a single token using linear layers. An action mapper extracts the hidden states of the action token from the final layer of the VLM, and transforms these to match the dimensionality expected by the pretrained action decoder, and finally the action decoder generates the corresponding actions with 7 degrees of freedom (DoF).

cess rate across all task suites, outperforming the previous state-of-the-art VLA method by 11.8%. Notably, on the challenging LIBERO-LONG benchmark (Liu et al., 2023a), our method achieves a **93.5%** success rate, surpassing the small action model Seer (Tian et al., 2024) by 5.8% and exceeding the best previously reported VLA result by 24.5%. On the CALVIN ABC-D benchmark, our method achieves the best performance among VLA-based approaches. Specifically, on the first task of five-task sequence, it achieves a success rate of **92.5%**, outperforming Seer (88.4%). We further evaluate VITA-VLA in 200 real-world trials across five manipulation tasks spanning four canonical operations—Pick, Place, Close, and Stack—using the ALOHA robot. Our method consistently outperforms Seer, achieving a **82.0%** average success rate (17.0% improvement).

## 2 RELATED WORK

**Small Action Models.** Small action models in language-conditioned robot manipulation typically adopt lightweight architectures with limited parameters, making them suitable for relatively simple tasks. These models often use a pretrained CLIP (Radford et al., 2021) text encoder to process language input and vision encoders such as CLIP or SigLIP (Zhai et al., 2023) to extract image features. In addition, a dedicated state encoder processes the robot states, and the combined multimodal inputs are passed to a high-capacity policy network to predict the corresponding actions. RT-1 (Brohan et al., 2022) uses the Universal Sentence Encoder to process text and a pretrained EfficientNet-B3 (Tan & Le, 2019) to encode images. The action information is discretized into 256 bins, and the image and text tokens are concatenated and fed into an 8-layer decoder-only Transformer, which generates action tokens end-to-end. Seer (Tian et al., 2024) uses MAE (He et al., 2022) to process visual input, the CLIP text encoder for text processing, and an MLP to encode state information. The action token is treated as a learnable query, which is decoded to generate action. RDT-1b (Liu et al., 2024) employs T5 as the text tokenizer, SigLIP as the image encoder, and a Diffusion Transformer as the core architecture. It generates actions through an iterative denoising process. Although these models demonstrate strong performance on short instruction-following tasks, they often struggle to generalize in complex or long-horizon tasks. This limitation arises primarily from the restricted capacity of their text and vision encoders. To address this, our method adopts the powerful VITA-1.5-7B (Fu et al., 2025) as backbone, a 7B-parameter VLM trained on large-scale, diverse datasets. This foundation enables significantly improved instruction comprehension, visual grounding, and long-horizon planning. This makes our approach more scalable, generalizable, and robust for robotic manipulation across varied environments.

**Large Vision-Language Action Models.** Recent progress in VLMs (Liu et al., 2023b; Achiam et al., 2023; Team et al., 2023; Bai et al., 2023) has advanced the development of VLA models for robotic control. RT-2 (Zitkovich et al., 2023) is the first to convert actions into discrete tokens and perform end-to-end autoregressive training using PaLM-E (12B) as the backbone. It is trained on 130k human-teleoperated demonstrations collected over 17 months. OpenVLA (Kim et al., 2024) extends this approach with an open-source 7B LLaMA model, using SigLIP and DINOv2 as vision encoders and trained on 970k episodes from the Open X-Embodiment dataset. Different from above, $\pi_0$ (Black et al., 2024) formulates action generation as a denoising process. It combines Gemma-2B as the VLM with a diffusion-based action expert, jointly trained over 68 dexterous tasks from diverse robot embodiments. GR00T (Bjorck et al., 2025) uses Eagle-2 (1.34B) as the VLM backbone and injects its vision-language features into a diffusion transformer via cross-attention, generating actions conditioned on the current state and noised actions through denoising. The entire model is trained on a dataset of 780k simulated trajectories. Despite their scale, these models require large datasets, long training times, and significant computational resources. In contrast, our approach provides a more efficient and scalable alternative. Instead of training a full VLA model from scratch, we introduce a two-stage distillation framework. In the first stage, we explicitly align the action representation spaces of a pretrained VLM and a small action model through lightweight representation matching, updating only a small subset of parameters to reduce computational cost. In the second stage, we directly attach the pretrained action decoder from the small action model to the VLM, forming a new VLA system. We then fine-tune the language model, state encoder, and the entire action module—including the action mapper and decoder—allowing the model to integrate the advanced capabilities of the VLM with the low-level control precision of the small action model, while remaining more efficient than training a VLA model from scratch.

## 3 METHOD

### 3.1 PROBLEM FORMULATION

We consider a robotic manipulation task formulated as a Markov Decision Process. Given a dataset $\mathcal{D} = \{(x_i, l_i, s_i, a_i)\}_{i=1}^{N}$ of $N$ demonstrations, each tuple consists of visual observation $x_i \in \mathcal{X}$, language instruction $l_i \in \mathcal{L}$, robot state $s_i \in \mathcal{S} \subseteq \mathbb{R}^{d_s=7}$ (6 degrees of freedom (DoF) arm state and 1 dimension gripper width), and a corresponding action $a_i \in \mathcal{A} \subseteq \mathbb{R}^{d_a=7}$ (comprising $a_i^{\mathrm{arm}}$ for the 6-DoF arm action and 1 dimension $a_i^{\mathrm{grip}}$ for the gripper width). Our objective is to learn a policy

$$\pi_\theta : \mathcal{X} \times \mathcal{L} \times \mathcal{S} \to \mathcal{A},$$

that predicts executable actions $\hat{a} = \pi_\theta(x, l, s)$ conditioned on multimodal inputs.

We further assume access to a pretrained small action model $\pi_{\mathrm{expert}} : \mathcal{X} \times \mathcal{L} \times \mathcal{S} \to \mathcal{A}$ that has been trained on the same dataset $\mathcal{D}$, which demonstrates strong performance on robotic manipulation tasks. Our goal is to distill the action modeling capability of $\pi_{\mathrm{expert}}$ into a vision-language model $f_\phi$, creating a unified VLA model that combines the visual-linguistic understanding and generalizability of VLMs with the precise action prediction of specialized action models.

### 3.2 OVERALL ARCHITECTURE

The overall architecture of our model is illustrated in Figure 2. Our architecture extends the pretrained VITA-1.5-7B model with minimal modifications to enable action prediction while preserving its vision-language capabilities. The backbone consists of three primary components: a vision encoder (InternViT-300M), a connector (3-layer MLP), and a language model (Qwen-2.5-7B). This backbone has been fine-tuned on large-scale open-source data covering a wide range of scenarios, demonstrating strong capabilities in image understanding and complex instruction following.

**State Input.** To incorporate robot state information, we initially explore concatenating raw state values as text tokens, but this approach fails because numerical values are poorly represented in the VLM's token vocabulary and the model struggles to interpret these values accurately. Thus, we design a dedicated state encoder. Specifically, the 6-DoF arm state and the 2-dimensional gripper state (one-hot encoding of the 1 dimension gripper width) are first encoded separately by two linear layers. Their outputs are then concatenated and passed through an additional linear layer, which projects the combined representation into the same dimension as the text tokens. This design allows the model to effectively integrate and attend to structured state information.

**Action Token.** We define a learnable *action token* that acts as a query during training and inference. This token is appended to the input sequence and is responsible for attending to the multimodal

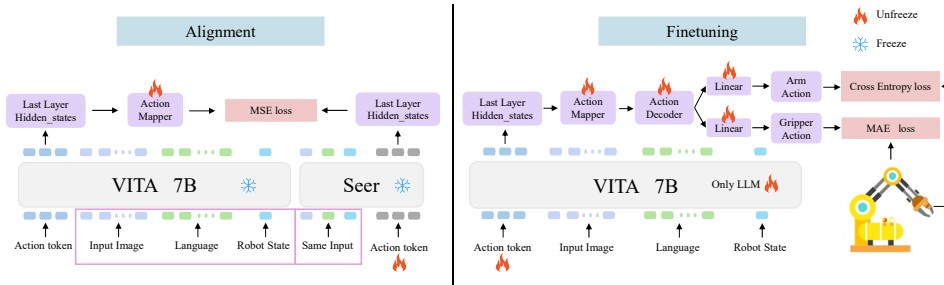

Figure 3: **Training Strategy.** Our training strategy comprises two stages. In the alignment stage, we train the action mapper, action tokens, and state encoder to bridge the gap between the action output spaces of the VLM and the small action model, updating only 30 million parameters while achieving improved fine-tuning outcomes. In the fine-tuning stage, we then perform end-to-end optimization of the entire model to further enhance overall performance.

context to produce action representations. To predict three future steps, the same token is repeated three times. Since consecutive actions are highly correlated and usually differ minimally, a single shared token is sufficient to capture temporal continuity. We also observed that assigning independent action tokens to each step does not yield additional benefits in our setting.

**Input.** Each training sample comprises 13 time steps. This setting follows the small action model, which is trained on 13-step trajectories, ensuring consistent temporal structure during alignment. At each step, the model receives a structured multimodal sequence: image tokens from two views (static and wrist cameras), text tokens (instruction), one state token, and three action tokens. Each image ($200 \times 200$) is encoded into 49 visual tokens, yielding $n = 98$ image tokens per step. Let $m$ be the number of instruction tokens; the complete per-step input is:

$$[\text{img}_1] \dots [\text{img}_n] \, [\text{text}_1] \dots [\text{text}_m] \, [\text{state}] \, [\text{act}_1] \, [\text{act}_2] \, [\text{act}_3]. \tag{1}$$

Formally, we denote the complete set of inputs as $x$ (image tokens), $t$ (text tokens), $s$ (state token), and $a_q$ (action token). The model $\pi$ jointly processes $\{x, t, s, a_q\}$ and the action embeddings are obtained by extracting the final-layer hidden states of $a_q$:

$$a_h = \pi_{\text{last}}(a_q \mid x, t, s). \tag{2}$$

**Action Mapper, Decoder and Output.** We employ a lightweight three-layer MLP as the *action mapper* $M$ to project $a_h$ into the input space expected by the pretrained action decoder. The first layer transforms the feature dimension of $a_h$ to match the action space. The other two layers keep the same size and introduce sufficient nonlinearity. This architecture strikes a balance between model expressiveness and computational efficiency. The mapped features are then fed into the pretrained action decoder $D$, which is a fixed two-layer MLP reused from the small action model. After the first-stage alignment, the output space of the action mapper becomes well aligned with the decoder's expected input space, enabling effective integration. Together, the action mapper and decoder generate the final executable action $\hat{a} = D(M(a_h))$.

### 3.3 TRAINING STRATEGY

To equip our VLA model with action execution capabilities, we introduce a two-stage training strategy. The core idea is to transfer the action modeling abilities from a small action model to the VLA through alignment. In the first stage, we employ lightweight alignment to bridge the gap between the action representation spaces of the VLA and the small action model, enabling the reuse of the pretrained action decoder from the small action model. In the second stage, end-to-end fine-tuning is conducted to further enhance the VLA model's action modeling capabilities. This approach reduces training resources while maintaining high performance in action generation.

#### STAGE 1: ALIGNMENT

The first stage, illustrated on the left side of Figure 3, aims to align the hidden action representations between the VLA model and the small action model. Except for the learnable action query, both models receive identical inputs, including the images, text instruction, and robot state. Since both

models follow an autoregressive architecture, we extract the last-layer hidden states corresponding to the action tokens from each model for alignment. Due to the difference in feature dimensions, we use an *action mapper* to transform the dimension of the VLA's action token hidden states into that of the small action model. We then compute a mean squared error (MSE) loss between the mapped VLA hidden states and the corresponding hidden states from the small action model:

$$\mathcal{L}_{\text{align}} = \frac{1}{N} \sum_{i=1}^{N} \left\| M(a_h^{\text{VLA},i}) - a_h^{\text{Small},i} \right\|_2^2, \tag{3}$$

where $M(\cdot)$ denotes the action mapper, $N$ is the number of action tokens, $a_h^{\text{VLA}}$ is the VLA model's last-layer hidden state of the action token, and $a_h^{\text{Small}}$ is the corresponding hidden state of the small action model. In this stage, only the state encoder, action tokens, and action mapper are trained, comprising a total of approximately 30 million parameters. This lightweight configuration enables efficient and fast alignment while preparing the model for the subsequent end-to-end fine-tuning.

STAGE 2: END-TO-END FINE-TUNING

After the alignment stage, the VLA model is better positioned for end-to-end fine-tuning. We continue using the same data as in Stage 1—comprising action tokens, images, text instructions, and robot states, as illustrated on the right side of Figure 3. These inputs are processed by the VLA model to produce the last-layer hidden states corresponding to the action tokens.

As in Stage 1, we apply the pretrained *action mapper* to project hidden states into the action space of the small action model. We reuse the pretrained *action decoder* and *linear projection heads* from the small action model to generate executable actions. The decoder is a two-layer MLP, followed by linear layers for predicting 6-DoF arm actions and binary gripper actions. Arm actions are supervised with mean absolute error (MAE) loss, and gripper actions with binary cross-entropy (BCE) loss. The total loss is their weighted sum: $\mathcal{L}_{\text{total}} = \mathcal{L}_{\text{arm}} + \lambda \cdot \mathcal{L}_{\text{gripper}}$ where:

- $\mathcal{L}_{\text{arm}} = \frac{1}{T} \sum_{t=1}^{T} \|\hat{a}_t^{\text{arm}} - a_t^{\text{arm}}\|_1$ is the MAE loss for the 6-DoF arm action, computed across all predicted time steps $T$. Here, $\hat{a}_t^{\text{arm}} \in \mathbb{R}^6$ denotes the predicted continuous arm action, and $a_t^{\text{arm}} \in \mathbb{R}^6$ denotes the ground truth.

- $\mathcal{L}_{\text{gripper}} = -\frac{1}{T} \sum_{t=1}^{T} \left[ a_t^{\text{grip}} \log(\hat{a}_t^{\text{grip}}) + (1 - a_t^{\text{grip}}) \log(1 - \hat{a}_t^{\text{grip}}) \right]$ is the BCE loss for the gripper action, where $\hat{a}_t^{\text{grip}} \in [0, 1]$ denotes the predicted probability of the gripper being closed, and $a_t^{\text{grip}} \in \{0, 1\}$ denotes the ground truth label.

- $\lambda = 0.01$ is a scaling factor set according to the observed loss magnitudes, ensuring that the gripper loss and arm loss contribute comparably during training.

This combined loss guides the model to learn accurate continuous arm actions and reliable binary gripper actions, maintaining balance between the two goals. In this stage, we fine-tune the LLM, state encoder, learnable action queries, action mapper, and action decoder, enabling end-to-end integration of multimodal information for accurate action prediction.

**Choice of MSE and MAE.** We adopt different loss functions in the two stages to suit their distinct goals. During alignment, MSE penalizes large deviations between VLM and small model hidden states, making it suitable for representation matching. In fine-tuning, the objective shifts to predicting continuous control signals. We use MAE for 6-DoF arm actions, as it yields more stable optimization and is less sensitive to outliers. This combination balances representation alignment with reliable low-level action supervision.

## 4 EXPERIMENTS

To validate the effectiveness of our model architecture and training strategy, we conduct simulation experiments on the CALVIN ABC-D and LIBERO benchmarks, as well as real-world experiments using a robotic arm platform. Our evaluation aims to address the following three questions: 1) Can this architecture perform well across various environments and tasks? 2) Does the proposed distillation process lead to measurable performance gains? 3) Can the model be effectively deployed on real-world robotic platforms?

Table 1: **CALVIN ABC-D results.** We report the success rates computed over 1000 rollouts for each task, along with the average number of completed tasks required to solve five instructions consecutively (denoted as Avg. Len.). The results for small action models without a VLM are displayed in the first three rows. The remaining rows represent VLA models that include a VLM. The best result in each category is highlighted in **bold**.
[*]: SAM is Small Action Model. † indicates results reproduced by us.

| Method | Category | VLM | Task completed in a row | | | | | |
|---|---|---|---|---|---|---|---|---|
| | | | 1 | 2 | 3 | 4 | 5 | Avg. Len. ↑ |
| Susie | | - | 87.0 | 69.0 | 49.0 | 38.0 | 26.0 | 2.69 |
| GR-1 | SAM[*] | - | 85.4 | 71.2 | 59.6 | 49.7 | 40.1 | 3.06 |
| **Seer-Large†** | | **-** | **88.4** | **80.9** | **74.8** | **69.6** | **62.1** | **3.76** |
| 3D-VLA | | BLIP2-4B | 44.7 | 16.3 | 8.1 | 1.6 | 0.0 | 0.71 |
| OpenVLA | | Prismatic-7B | 62.8 | 18.3 | 5.8 | 1.8 | 1.0 | 0.90 |
| Roboflamingo | VLA | Flamingo-3B | 82.4 | 61.9 | 46.6 | 33.1 | 23.5 | 2.48 |
| Ours(only-ft) | | VITA1.5-7B | 86.0 | 73.2 | 60.4 | **49.4** | **39.8** | 3.08 |
| **Ours (two-stage)** | | **VITA1.5-7B** | **92.5** | **77.1** | **61.0** | 49.2 | 38.2 | **3.18** |

## 4.1 BENCHMARKS, BASELINES AND EXPERIMENT DETAILS

**Benchmarks.** We evaluate our model on two robotic manipulation benchmarks: CALVIN and LIBERO. CALVIN is a simulated benchmark comprising 34 tasks and 1,000 language instructions across four environments (A–D), each with different desk colors and object layouts. Following the ABC-D setting, models are trained on environments A, B, and C, and tested on the unseen environment D to assess generalization. LIBERO is a comprehensive lifelong learning benchmark with four task suites—Spatial, Object, Goal, and LONG—each suite contains 10 long-horizon tasks, evaluating different aspects of generalization in robotic manipulation.

**Baselines.** We compare our method with a diverse set of representative VLA baselines. For LIBERO, we include large-scale pretrained generalist models (OpenVLA (Kim et al., 2024), Octo (Team et al., 2024)) as well as architectures with enhanced grounding or reasoning capabilities, such as SpatialVLA (spatial information) (Qu et al., 2025), CoT-VLA (visual chain-of-thought) (Zhao et al., 2025), and $\pi_0$-Fast (flow-based method) (Black et al., 2024). For CALVIN ABC-D, we evaluate both small action models—Susie (diffusion-based subgoal planning) (Black et al., 2023) and GR-1 (video pretraining) (Wu et al., 2023)—and VLA-based methods including Roboflamingo (Li et al., 2023), 3D-VLA (Zhen et al., 2024), and OpenVLA. Seer (Tian et al., 2024) is used in both benchmarks and also serves as the distillation teacher.

**Experiment Details.** Our experiments involve three distinct training settings: 1) Two-stage: a two-stage training strategy, where the model is first aligned and then fine-tuned. 2) Only-finetune: no alignment stage is performed. Instead, we directly attach the pretrained action module from the small action model to our VLM, and then perform only the fine-tuning procedure from the two-stage protocol to train the combined model. 3) Freeze-vlm: directly integrate the pretrained action token, state encoder, and action module weights (from the first strategy) into the VLM, but do not update the VLM parameters during training, to test whether a strong VLA could be achieved without tuning the VLM. In all experiments, we use only language-conditioned data, corresponding to 58% of the full training set in CALVIN ABC-D. Additional hyperparameter settings are provided in Appendix A.1.

## 4.2 MAIN RESULTS

**Zero-Shot Generalization to Unseen Environments.** Table 1 presents the results on the CALVIN ABC-D benchmark. Our model achieves the highest performance among existing VLA models when trained on environments A, B, and C and evaluated in the unseen environment D, demonstrating strong generalization capability to unseen scenes. Although our model achieves a higher success rate than Seer-Large on Task 1 (92.5% vs. 88.4%), it yields a lower average task success length. We attribute this to the model's sensitivity to environmental changes during task transitions, which may lead the VLM to misinterpret context or lose consistency across subtasks. This highlights a potential area for improving temporal robustness in long-horizon manipulation.

**Long-Horizon Planning and Complex Instruction Execution.** As shown in Table 3, our model achieves a 5.8% improvement in average success rate on the LIBERO-LONG benchmark compared to the Seer-Large model. We attribute this gain to the integration of a VLM, which improves the

Table 2: **LIBERO results of different VLA models.** We present the success rates of various VLA models. To ensure fair comparison, we report the average success rates over 500 episodes, following the evaluation protocol used in CoT-VLA (Zhao et al., 2025). The best result in each category is highlighted in **bold**. Our model achieves state-of-the-art performance across all tasks, demonstrating exceptional long-horizon execution capabilities. Notably, it raises the average success rate to 97.3%, in the LIBERO-LONG task, it improves the previous state-of-the-art by 24.5%.

| Method | SPATIAL | OBJECT | GOAL | LONG | Average |
|---|---|---|---|---|---|
| Octo | 78.9% | 85.7% | 84.6% | 51.1% | 75.1% |
| OpenVLA | 84.9% | 88.4% | 79.2% | 53.7% | 76.5% |
| SpatialVLA | 88.2% | 89.9% | 78.6% | 55.5% | 78.1% |
| CoT-VLA | 87.5% | 91.6% | 87.6% | 69.0% | 81.1% |
| $\pi_0$-FAST | 96.4% | 96.8% | 88.6% | 60.2% | 85.5% |
| Ours(two-stage) | **98.0 %** | **99.8 %** | **97.9 %** | **93.5%** | **97.3%** |

Table 3: **LIBERO-LONG results across different tasks.** For each task, we report the average success rate over 20 rollouts, following the evaluation protocol used in Seer (Tian et al., 2024). The metric "Avg.Success" denotes the average success rate across all ten tasks. The best results are highlighted in **bold**. Our model achieves the best performance on LIBERO-LONG. It demonstrates a 5.8% improvement over Seer-Large and a 1% improvement over the only-finetune strategy, showcasing its proficiency in executing tasks that require long-horizon planning. Detailed task information is provided in Appendix A.3.

| Method | Avg. Success ↑ | Task 1 | Task 2 | Task 3 | Task 4 | Task 5 | Task 6 | Task 7 | Task 8 | Task 9 | Task 10 |
|---|---|---|---|---|---|---|---|---|---|---|---|
| MPI | 77.3 | 66.6 | 86.6 | 96.6 | 95.0 | 83.3 | 83.3 | 56.6 | 86.6 | 40.0 | 78.3 |
| OpenVLA | 54.0 | 35.0 | 95.0 | 65.0 | 45.0 | 40.0 | 80.0 | 60.0 | 45.0 | 20.0 | 55.0 |
| Seer-large | 87.7 | 91.7 | 90.0 | 98.3 | **100.0** | 91.7 | 93.3 | 85.0 | 88.3 | 61.7 | 71.7 |
| Ours(only-ft) | 92.5 | 91.7 | 100.0 | 98.3 | 98.3 | 98.3 | 90.0 | 90.0 | **91.7** | **86.7** | 80.0 |
| Ours(two-stage) | **93.5** | **100.0** | **100.0** | **100.0** | **100.0** | **100.0** | **95.0** | **95.0** | 80.0 | 75.0 | **90.0** |

model's ability to understand and execute complex instructions and to more effectively process multimodal (language, visual, states) inputs. Furthermore, as shown in Table 2, our model outperforms all other VLA models on the LIBERO benchmark. Specifically, compared with the previous best success rate of 69.0%, our model improves by 24.5% on LIBERO-LONG and achieves an overall average success rate increase of 11.8% across all tasks.

**Effectiveness of the Two-Stage Training Strategy.** As shown in Tables 1 and 3, the two-stage trained model consistently outperforms the only-finetune baseline, achieving a 6.5% increase in task 1 success rate on the CALVIN ABC-D benchmark and a 1% improvement in the average success rate on the LIBERO-LONG benchmark. These results clearly demonstrate the effectiveness of the proposed two-stage training framework, further confirming that aligning the action representation spaces prior to fine-tuning leads to superior performance in robotic manipulation tasks.

**Direct Integration of Action Module Weights with Frozen VLM.** In this approach, we directly incorporate the pretrained action token, state encoder, and action module weights obtained from the two-stage training pipeline into the original VLM, while keeping the VLM parameters frozen during end-to-end fine-tuning. This setup aims to assess whether a robust VLA model can be achieved solely by adapting the action module, without updating the VLM. Evaluation on the CALVIN ABC-D benchmark reveals a first-task success rate of only 45.3%, indicating that freezing the VLM component significantly constrains overall performance. These results highlight the necessity of fine-tuning the VLM to equip it with action-execution capabilities.

## 4.3 REAL-WORLD EVALUATION

**Real World Settings.** We design five real-world tasks to comprehensively evaluate the model's capabilities, covering four canonical robotic operations: Pick, Place, Close, and Stack. Our real-world experiments are conducted on the ALOHA platform, where the arm is precisely controlled by six joint angles and the gripper is controlled through its opening width. To enable a fair evaluation, we manually collect 500 high-quality demonstration trajectories (100 per task) across a wide range of scenarios. Both the Seer model and the proposed model are trained on this dataset, with the latter employing two distinct training strategies for systematic comparison. For evaluation, we report the

| Close Drawer | Stack Cups | Stack Blocks | Pick Place Sponge | Pick Place Block |
|---|---|---|---|---|

Figure 4: **Real-world Tasks.** To evaluate the model in real-world settings, we formulate five tasks that span four canonical operations: Pick, Place, Close, and Stack.

task success rate averaged over 40 independent real-world roll-out trials, providing a robust measure of performance. The natural language instructions for the five real-world tasks are as follows: (1) close the drawer, (2) stack the orange cup on top of the green cup, (3) stack the red block on top of the yellow block, (4) pick up the sponge and put it into the basket, and (5) pick up the red block and put it into the basket. The corresponding real-world scenarios are illustrated in Figure 5.

**Detailed Results.** The detailed experimental results are summarized in Table 4. Our two-stage model achieves the best performance across all tasks, surpassing both the Seer model and the fine-tuned baseline, which demonstrates the effectiveness of our strategy in real-world settings. For relatively short and simple tasks such as *Close Drawer*, all three models achieve comparable and satisfactory performance. However, in longer-horizon tasks such as *Pick and Place*, our model exhibits clear advantages. In particular, in the *Pick Place Block* task, accurate prediction of the gripper's opening width is crucial for successful execution, and our model demonstrates superior precision compared with the baselines. For more complex tasks such as *Stack Cups* and *Stack Blocks*, success requires both predicting fine-grained gripper widths and accurately identifying the correct opening positions to achieve stable stacking. These tasks further demand real-time perception of positional changes and a stronger understanding of visual information, where our VLA model significantly outperforms the Seer model. Additional implementation details are provided in Appendix A.2, and the real-world deployment results are presented in Appendix A.4.

Table 4: **Real-world Results.** We report the average success rate of each task over 40 rollouts. Our model achieves the best results across all tasks.

| Method | Success Rate (%) ↑ | | | | | |
|---|---|---|---|---|---|---|
| | Close Drawer | Stack Cups | Stack Blocks | Pick Place Sponge | Pick Place block | Avg. Score |
| Seer | 87.5 | 32.5 | 60.0 | 75.0 | 70.0 | 65.0 |
| Ours (only-ft) | 95.0 | 52.5 | 75.0 | 85.0 | 87.5 | 79.0 |
| Ours(two-stage) | **97.5** | **52.5** | **80.0** | **87.5** | **92.5** | **82.0** |

## 5 CONCLUSION

In this work, we propose a simple yet effective VLA model architecture and validate its effectiveness. By combining a pretrained VLM with a small action model, we enable the VLM to acquire action-execution capabilities through lightweight training while largely preserving its original structure. To train our model efficiently, we further introduce a two-stage distillation framework for transferring action-generation capabilities from small action models to the VLA model. The first stage aligns action representation spaces via lightweight representation matching, substantially reducing training complexity. The second stage selectively fine-tunes the language model, state encoder, and action modules, allowing the VLA to integrate the advanced ability of large-scale VLMs with the precise action-generation ability of small action model. Experimental results show competitive performance on CALVIN ABC-D and state-of-the-art results on LIBERO. Moreover, real-world robotic experiments confirm the practical applicability of our approach.

**Limitation and Future Work.** Despite its simplicity and effectiveness, VITA-VLA depends on pretrained models, which constrains its application in domains lacking appropriate action experts. Additionally, it exhibits slower inference compared to small models. Future efforts will aim to alleviate these limitations by enhancing efficiency and reducing reliance on pretrained models.

## REPRODUCIBILITY STATEMENT

To ensure reproducibility, we release our source code and tutorials at https://anonymous.4open.science/r/VLA-Model-C16E.

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

## A APPENDIX

### A.1 IMPLEMENTATION DETAILS

**Training Hyperparameters.** In our training process, we employ DeepSpeed's ZeRO-2 stage to efficiently train our model. This approach optimizes memory usage and accelerates training, making it suitable for handling large-scale datasets. The specific training hyperparameters used in our experiments are detailed in Table 5.

**Model Hyperparameters.** Since we conduct experiments on different datasets, we use different pretrained Seer models for alignment. This leads to variations in the hyperparameters of our model across datasets, as shown in Table 6.

Table 5: Training Hyperparameters

| Hyperparameters | Alignment | Finetuning |
|---|---|---|
| batch size | 8 | 4 |
| gradient accumulation steps | 4 | 4 |
| learning rate | 1e-4 | 1e-4 |
| optimizer | AdamW | AdamW |
| learning rate schedule | cosine decay | cosine decay |
| warmup epochs | 1 | 2 |
| training epochs | 3 | 2 |
| arm loss ratio | - | 1.0 |
| gripper loss ratio | - | 0.01 |
| CALVIN max history length | 10 | 10 |
| LIBERO max history length | 7 | 7 |
| future action prediction | 3 | 3 |

Table 6: Model Hyperparameters

| | In dim | Out dim |
|---|---|---|
| Action token | - | 3584 |
| Arm action encoder(Linear) | 6-DoF | 3584 |
| Gripper action encoder(Linear) | 2 | 3584 |
| State projector(Linear) | 7168 | 3584 |
| Action mapper(3 layer MLP) | 3584 | 1024(CALVIN)/384(LIBERO) |
| Action decoder(2 layer MLP) | 1024(CALVIN)/384(LIBERO) | 512(CALVIN)/192(LIBERO) |
| Arm action decoder(Linear) | 512(CALVIN)/192(LIBERO) | 6-DoF(CALVIN)/1(LIBERO) |
| LLM(28 layer) | 3584 | 3584 |
| Vision encoder | $200 \times 200$ | $49 \times 3584$ |

**Image Resolution.** The image resolutions for the CALVIN datasets are $200 \times 200$ and $84 \times 84$, while the resolution for LIBERO is $128 \times 128$. To unify resolution across datasets, we resize all images to $200 \times 200$, resulting in 49 tokens per image. Since most training data for the VLM are $224 \times 224$ images, we also resize images to $224 \times 224$ for comparison. However, due to the originally low resolution of the images, upscaling to $224 \times 224$ causes the VLM to struggle even with basic image understanding tasks. After training, performance with $224 \times 224$ input is actually worse than with the $200 \times 200$ setting.

**Action Mapper Architecture.** We use the action mapper to transform the hidden states of the action tokens into the dimensionality expected by the pretrained action decoder. The action mapper can adopt different architectures, such as MLPs, transformer-based architectures, decoder-only, or encoder-only architectures. We compare MLP and decoder-based implementations and find the performance difference to be negligible. Since the MLP is simpler and aligns with Occam's razor, we adopt the MLP as our action mapper.

## A.2 REAL ROBOT EXPERIMENT SETTINGS

Our real-world robotic platform is illustrated in Fig. 5. The setup consists of two cameras: a base-mounted Intel RealSense D435i RGB-D camera with a resolution of 1280×720, and a gripper-mounted Dabai DCW depth camera with a resolution of 640×480, providing complementary viewpoints for perception. The robot itself is a PiPer arm with six actuated joints, controlled in radians, equipped with a Songling parallel gripper whose opening width is directly commanded for grasping. This combination allows both global scene observation and fine-grained local perception at the end-effector, facilitating precise manipulation. Demonstration data were collected via teleoperation, and the same hardware was used for inference. The platform is powered by a workstation with a single GPU, on which our model runs at approximately 0.15s per inference step (about 6–7 Hz). For comparison, the Seer model achieves about 0.05s per inference (roughly 20 Hz), highlighting a trade-off between inference speed and action accuracy.

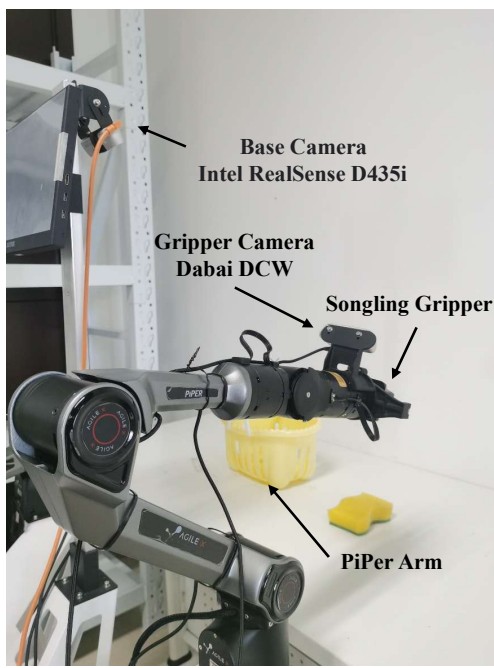

Figure 5: **Real robot setup.** The platform consists of a PiPer robotic arm with a Songling gripper, equipped with two complementary cameras: an Intel RealSense D435i base camera (1280×720) and a Dabai DCW gripper-mounted depth camera (640×480).

## A.3 DETAILED RESULTS OF LIBERO EXPERIMENTS

We evaluate on all ten tasks from the LIBERO-LONG benchmark, the result is like Table 3: task1: Put soup and box in basket, task2: Put box and butter in basket, task3: Turn on stove and put pot, task4: Put bowl in drawer and close it, task5: Put mugs on left and right plates, task6: Pick book and place it in back, task7: Put mug on plate and put pudding to right, task8: Put soup and sauce in basket, task9: Put both pots on stove, and task10: Put mug in microwave and close it.

We evaluate our model on the LIBERO benchmark. To present the results more clearly and intuitively, we sample evaluation data and visualize the process on LIBERO, as illustrated in Figure 6.

We also report the success rates for different tasks across various benchmarks, as shown in Table 7. The results show that our model performs consistently well across all tasks, demonstrating its effectiveness in handling long-horizon and complex tasks.

## A.4 REAL-WORLD DEPLOYMENT

We further validate our method by deploying the model on the Aloha robotic arm in real-world scenarios. Specifically, we evaluate it across five distinct manipulation tasks, which can be categorized into four fundamental operation types—*pick*, *place*, *stack*, and *close*—all of which require stable and precise action outputs for successful completion. As shown in Fig. 7, our model achieves high success rates and stable performance on all tasks, demonstrating strong generalization ability and robustness when transferred from simulation to reality.

## THE USE OF LLMS

We use the GPT-4o model to assist with grammar correction and language refinement during the writing of this paper. We thank the developers of GPT-4o for providing such a helpful tool.

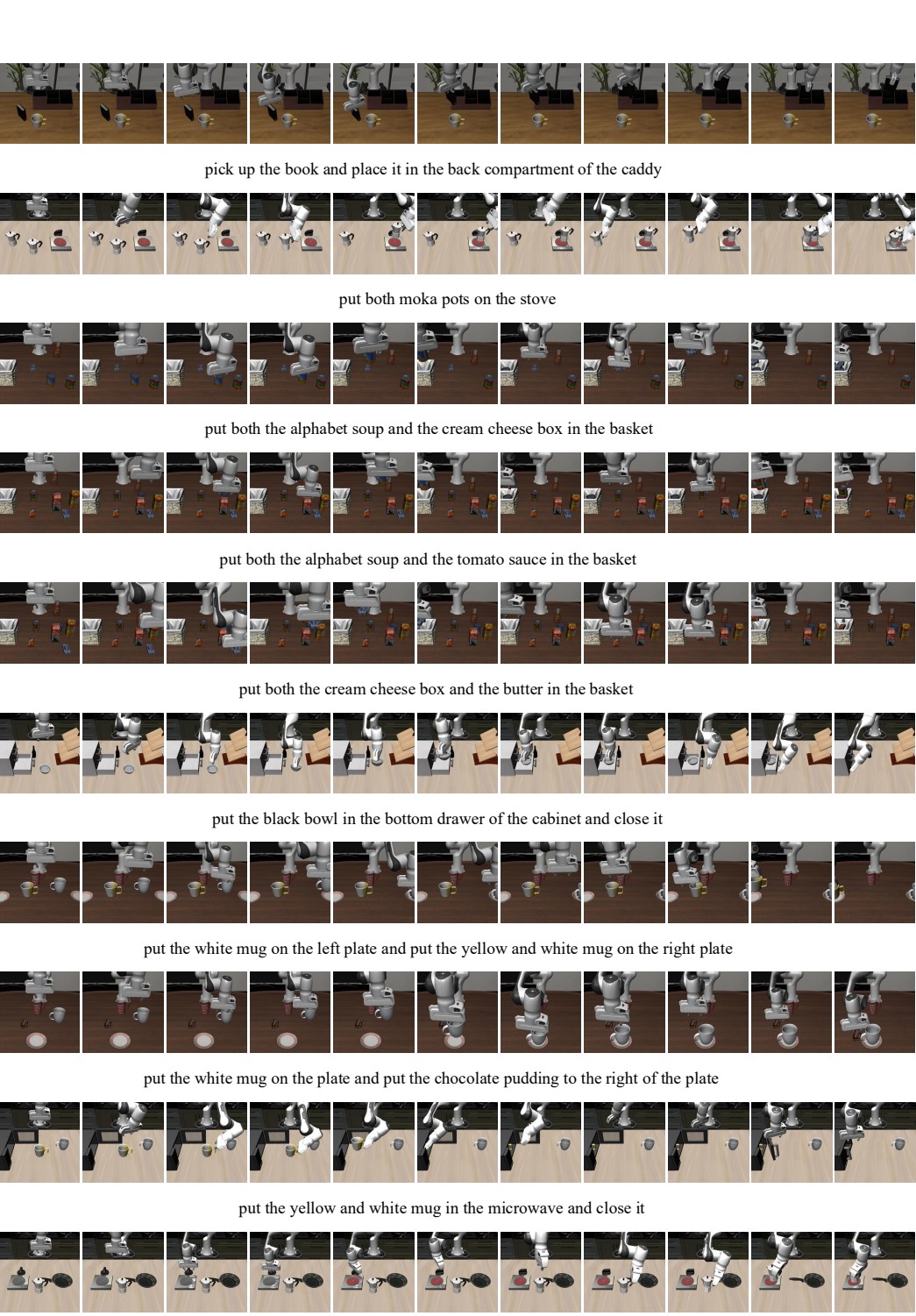

pick up the book and place it in the back compartment of the caddy

put both moka pots on the stove

put both the alphabet soup and the cream cheese box in the basket

put both the alphabet soup and the tomato sauce in the basket

put both the cream cheese box and the butter in the basket

put the black bowl in the bottom drawer of the cabinet and close it

put the white mug on the left plate and put the yellow and white mug on the right plate

put the white mug on the plate and put the chocolate pudding to the right of the plate

put the yellow and white mug in the microwave and close it

turn on the stove and put the moka pot on it

Figure 6: LIBERO-10 Visualization.

Table 7: Task Success Rates of LIBERO Benchmarks

(a) LIBERO-Spatial

| Task ID | Task Name | Accuracy (%) |
|---------|-----------|--------------|
| 0 | Pick_black_bowl_between_plate_and_ramekin_place_on_plate | 100.0 |
| 1 | Pick_black_bowl_next_to_ramekin_place_on_plate | 100.0 |
| 2 | Pick_black_bowl_from_table_center_place_on_plate | 100.0 |
| 3 | Pick_black_bowl_on_cookie_box_place_on_plate | 100.0 |
| 4 | Pick_black_bowl_in_top_drawer_of_wooden_cabinet_place_on_plate | 100.0 |
| 5 | Pick_black_bowl_on_ramekin_place_on_plate | 96.0 |
| 6 | Pick_black_bowl_next_to_cookie_box_place_on_plate | 100.0 |
| 7 | Pick_black_bowl_on_stove_place_on_plate | 100.0 |
| 8 | Pick_black_bowl_next_to_plate_place_on_plate | 94.0 |
| 9 | Pick_black_bowl_on_wooden_cabinet_place_on_plate | 90.0 |

(b) LIBERO-Goal

| Task ID | Task Name | Accuracy (%) |
|---------|-----------|--------------|
| 0 | Pick_up_the_alphabet_soup_and_place_it_in_the_basket | 100.0 |
| 1 | Pick_up_the_cream_cheese_and_place_it_in_the_basket | 100.0 |
| 2 | Pick_up_the_salad_dressing_and_place_it_in_the_basket | 100.0 |
| 3 | Pick_up_the_bbq_sauce_and_place_it_in_the_basket | 100.0 |
| 4 | Pick_up_the_ketchup_and_place_it_in_the_basket | 100.0 |
| 5 | Pick_up_the_tomato_sauce_and_place_it_in_the_basket | 100.0 |
| 6 | Pick_up_the_butter_and_place_it_in_the_basket | 100.0 |
| 7 | Pick_up_the_milk_and_place_it_in_the_basket | 100.0 |
| 8 | Pick_up_the_chocolate_pudding_and_place_it_in_the_basket | 97.9 |
| 9 | Pick_up_the_orange_juice_and_place_it_in_the_basket | 100.0 |

(c) LIBERO-Object

| Task ID | Task Name | Accuracy (%) |
|---------|-----------|--------------|
| 0 | open_the_middle_drawer_of_the_cabinet | 100.0 |
| 1 | put_the_bowl_on_the_stove | 100.0 |
| 2 | put_the_wine_bottle_on_top_of_the_cabinet | 87.5 |
| 3 | open_the_top_drawer_and_put_the_bowl_inside | 100.0 |
| 4 | put_the_bowl_on_top_of_the_cabinet | 100.0 |
| 5 | push_the_plate_to_the_front_of_the_stove | 97.9 |
| 6 | put_the_cream_cheese_in_the_bowl | 100.0 |
| 7 | turn_on_the_stove | 100.0 |
| 8 | put_the_bowl_on_the_plate | 95.8 |
| 9 | put_the_wine_bottle_on_the_rack | 97.9 |

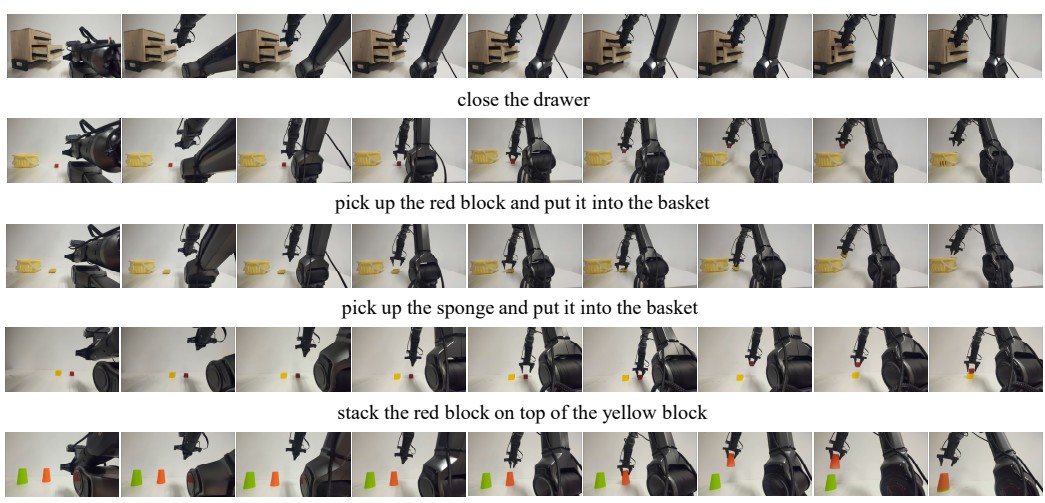

close the drawer

pick up the red block and put it into the basket

pick up the sponge and put it into the basket

stack the red block on top of the yellow block

stack the orange cup on top of the green cup

Figure 7: Five real-world tasks on the Aloha robotic arm.

