# OpenReview forum: "VITA-VLA: Efficiently Teaching Vision-Language Models to Act via Action Expert Distillation"
_ICLR.cc/2026/Conference — ICLR 2026 Conference Withdrawn Submission_

### Official Review · Reviewer_EmRx · 2025-10-30

**Soundness:** 3
**Presentation:** 2
**Contribution:** 2
**Rating:** 4
**Confidence:** 5

**Summary:**

This paper proposes VITA-VLA, a new framework that enables pretrained vision-language models (VLMs) to perform robotic manipulation by distilling knowledge from a small action model. Training proceeds in two stages: first, representation alignment maps the VLM’s hidden states into the action space of the expert model; second, selective fine-tuning integrates multimodal reasoning and precise action prediction. Experiments on the CALVIN and LIBERO benchmarks demonstrate outstanding performance.

**Strengths:**

S1. It is a good idea that the authors leverage distillation and VLM-alignment-like methods for VLA training. However, the choice of models and training strategy currently does not make much sense.

S2. The authors have conducted extensive simulation experiments.

**Weaknesses:**

**W1. Has the author investigated whether incorporating robot state into the VLA affects the trajectory-level generalization?** Based on the reviewer’s experimental experience, adding robot state indeed improves in-domain performance, but it tends to limit generalization in out-of-domain settings. Moreover, including robot state is a fairly basic operation and should not be presented as a contribution.

**W2. The motivation is not very clear.** Distilling from another action expert is a good idea, but why specifically choose Seer? Would distilling from such a specific action policy constrain the original VLM’s generalization and reasoning abilities? In other words, the process of distillation and the use of an Action Mapper may weaken the representation power of a large-scale foundation model. If the goal is to reduce training cost, why not distill from a large-scale pretrained VLA model instead?

**W3. Pretraining on large embodied datasets is not only intended to enhance manipulation performance on downstream tasks, but also to improve generalization ability.** The author could consider conducting generalization experiments on simpler environments such as Simpler-env [a].
[a] Evaluating Real-World Robot Manipulation Policies in Simulation

**W4. Why are the real-world experiments only compared against Seer?** The comparison should include more VLA models, such as π0-FAST, CogACT [b], and RDT-1B, to make the evaluation more comprehensive.
[b] CogACT: A Foundational Vision-Language-Action Model for Synergizing Cognition and Action in Robotic Manipulation

**Questions:**

The motivation and corresponding methodology should be revised. The idea itself makes sense, but distilling from a small policy specialized in a particular domain fundamentally contradicts the core principle of designing a VLA, which is to enable broad generalization.

---

### Official Review · Reviewer_udfZ · 2025-10-31

**Soundness:** 4
**Presentation:** 3
**Contribution:** 3
**Rating:** 4
**Confidence:** 3

**Summary:**

The paper proposes a two-stage distillation framework to transfer the capabilities of an action expert model into a pretrained vision-language model. It consists of two stages for training: Stage 1 aligns hidden representations between the VLM and the expert; stage 2 fine-tunes selected components for executable action generation. Experiments on CALVIN, LIBERO, and real-world tasks show its strong results.

**Strengths:**

- The framework introduces a few additional parameters, which drastically reduce training costs compared to fully end-to-end VLA training.
- The representation alignment and fine-tuning stages are conceptually clear and empirically justified. Ablation studies confirm that the second stage makes meaningful contributions to the final performance.
- The model achieves superior results across multiple simulation benchmarks and real-world experiments, outperforming existing approaches.
- The paper provides detailed code, training configurations, dataset descriptions, and hyperparameters, supporting reproducibility.

**Weaknesses:**

- While the proposed distillation framework reduces the overall training cost of the model, the training of the action expert itself is nontrivial. Moreover, the two-stage optimization pipeline adds additional engineering complexity and requires careful hyperparameter tuning.
- The experiments appear to rely on a single type of action expert throughout the paper. No ablation studies or cross-expert comparisons demonstrate that the proposed framework generalizes to other types of expert models. Similarly, it remains unclear whether the framework can adapt well to other vision-language backbones beyond those mentioned.
- The real-world tasks used for evaluation are relatively simple and short-horizon, which limits the demonstration of the framework’s ability to handle temporally extended reasoning and planning. Moreover, Table 4 shows that the representation alignment stage does not bring a substantial performance boost, suggesting that its contribution may be limited. It would be important to evaluate the framework on more challenging long-horizon real-world tasks to determine whether the alignment phase truly helps in complex sequential decision-making. If not, its necessity in the overall pipeline should be reconsidered.

**Questions:**

- What is the total number of trainable parameters in VITA-VLA compared to the standalone action expert and VLA baselines? Such numbers would help substantiate the claimed efficiency of the framework.
- Could the authors provide a quantitative comparison of the total training cost among the proposed two-stage framework, the action expert, and the baseline VLA models? Specifically: How many total GPU hours or FLOPs are required for (a) the action expert training, (b) the Stage-1 representation alignment, and (c) the Stage-2 fine-tuning?
- How does the inference-time efficiency of VITA-VLA compare with standard VLA baselines?

---

### Official Review · Reviewer_Tftg · 2025-11-01

**Soundness:** 3
**Presentation:** 3
**Contribution:** 3
**Rating:** 4
**Confidence:** 4

**Summary:**

This paper introduces VITA-VLA, a cost-effective framework for equipping vision-language models (VLMs) with action-execution capabilities via action expert distillation. The method avoids expensive end-to-end training by adopting a two-stage distillation process: (1) a lightweight action mapper aligns VLM features with a pretrained small action model's action space, and (2) selective fine-tuning integrates multimodal inputs (via an action token and state encoder) for precise action generation. Retaining the original VLM structure while adding minimal components, VITA-VLA achieves good performance on robotics benchmarks

**Strengths:**

1. Compared to training large-scale VLA models from scratch, the proposed method significantly reduces computational overhead through an efficient knowledge distillation scheme.

2. By preserving the pretrained VLM architecture and introducing only minimal structural modifications, the approach enables seamless integration into real-world robotic systems, aligning well with practical deployment constraints.

3. The overall workflow — transferring knowledge from the action expert to the VLM via distillation — is conceptually clear, and the paper is well-written with good readability.

**Weaknesses:**

1. The paper claims that "this approach removes the need for expensive end-to-end pretraining..." which is a desired outcome of distillation; however, it does not adequately justify how this method differs from existing VLA distillation approaches or provide quantitative comparative analysis.

2. This paper compares with several VLA methods, but lacks the latest and state-of-the-art methods, such as pi0.5 and GR00Tn1.5, which are already open source.

3. These designed tasks appear relatively simple, raising concerns about potential performance differences across tasks of varying difficulty. For example, it remains unclear whether VITA-VLA can still maintain its advantages on tasks requiring complex reasoning.

4. Another issue is the lack of details on the computational resources required to train VITA-VLA (e.g., GPU hours, memory usage). These are critical metrics for distillation and efficiency-focused methods, yet they are not reported

**Questions:**

N/A

---

### Official Review · Reviewer_Ci9H · 2025-11-02

**Soundness:** 2
**Presentation:** 3
**Contribution:** 2
**Rating:** 4
**Confidence:** 4

**Summary:**

This paper proposes a Knowledge Distillation-based framework designed to endow pre-trained Vision-Language Models (VLMs) with action execution capabilities, thereby constructing Vision-Language-Action (VLA) models.

The core of this method lies in transferring action knowledge from a pre-trained, smaller action model, which avoids the need for expensive end-to-end training.

The specific implementation comprises two stages:

Lightweight Alignment: Mapping the VLM's hidden states to the action model's space to reuse its action decoder.

Selective Fine-tuning: Fine-tuning the Language Model, the State Encoder, and the Action Module to enable the system to integrate multi-modal inputs and generate precise actions.

**Strengths:**

The paper is well-written, with clear narration and a logically structured organization. It effectively conveys the proposed concepts and methods, making it easy to understand.

**Weaknesses:**

The paper claims that diffusion-based methods restrict the ability of vision-language models (VLMs) to represent the robot arm’s state but does not clearly specify the source or reasoning behind this conclusion. Meanwhile, the proposed method also connects a three-layer MLP as an Action Mapper after the VLM. Would this likewise impose similar constraints on the VLM’s capability to represent robot states?

Insufficient experimental comparison and scene diversity:
The real-world experiments lack comparisons with stronger baseline models, such as pi0/05, OpenVLA-OFT.
Moreover, the performance gap between the only-finetuned model (only-ft) and the two-stage model on real-world pick-and-place*tasks is minor, making it difficult to highlight the advantages of the proposed approach. It is recommended to include experiments in more complex scenarios (e.g., folding clothes) and further evaluate the model’s robustness under varying backgrounds and lighting conditions.

Model architecture and generalization could be improved:
The current Action Mapper and Action Decoder are both implemented as three-layer MLPs, which are relatively small and may limit the quality of action generation. It is suggested to conduct ablation studies with larger networks to assess how model capacity affects performance.
Furthermore, the paper does not clearly explain what specific advantages using a three-layer MLP as the Action Decoder has over alternatives such as the FAST Action De-Tokenizer in pi. Adding comparative analysis would strengthen the discussion.

**Questions:**

See weakness

---

### Note · Authors · 2025-11-14

I have read and agree with the venue's withdrawal policy on behalf of myself and my co-authors.